# Bacterial Nanocellulose Nitrates

**DOI:** 10.3390/nano9121694

**Published:** 2019-11-27

**Authors:** Vera V. Budaeva, Yulia A. Gismatulina, Galina F. Mironova, Ekaterina A. Skiba, Evgenia K. Gladysheva, Ekaterina I. Kashcheyeva, Olga V. Baibakova, Anna A. Korchagina, Nadezhda A. Shavyrkina, Dmitry S. Golubev, Nikolay V. Bychin, Igor N. Pavlov, Gennady V. Sakovich

**Affiliations:** 1Laboratory of Bioconversion, Institute for Problems of Chemical and Energetic Technologies, Siberian Branch of the Russian Academy of Sciences (IPCET SB RAS), Biysk 659322, Altai Krai, Russia; julja.gismatulina@rambler.ru (Y.A.G.); yur_galina@mail.ru (G.F.M.); eas08988@mail.ru (E.A.S.); evg-gladysheva@yandex.ru (E.K.G.); massl@mail.ru (E.I.K.); olka_baibakova@mail.ru (O.V.B.); yakusheva89_21.ru@mail.ru (A.A.K.); 32nadina@mail.ru (N.A.S.); reklatekoy@gmail.com (D.S.G.); nbych@ya.ru (N.V.B.); pawlow-in@mail.ru (I.N.P.);; 2Biysk Technological Institute, Polzunov Altai State Technical University, Biysk 659305, Altai Krai, Russia

**Keywords:** carbon nanomaterials, bacterial nanocellulose, *Medusomyces gisevii*, nitration, nitric-sulfuric acids, cellulose nitrates, surface morphology, scale-up

## Abstract

Bacterial nanocellulose (BNC) whose biosynthesis fully conforms to green chemistry principles arouses much interest of specialists in technical chemistry and materials science because of its specific properties, such as nanostructure, purity, thermal stability, reactivity, high crystallinity, etc. The functionalization of the BNC surface remains a priority research area of polymers. The present study was aimed at scaled production of an enlarged BNC sample and at synthesizing cellulose nitrate (CN) therefrom. Cyclic biosynthesis of BNC was run in a semisynthetic glucose medium of 10−72 L in volume by using the *Medusomyces gisevii* Sa-12 symbiont. The most representative BNC sample weighing 6800 g and having an α-cellulose content of 99% and a polymerization degree of 4000 was nitrated. The nitration of freeze-dried BNC was performed with sulfuric-nitric mixed acid. BNC was examined by scanning electron microscopy (SEM) and infrared spectroscopy (IR), and CN was explored to a fuller extent by SEM, IR, thermogravimetric analysis/differential scanning analysis (TGA/DTA) and ^13^C nuclear magnetic resonance (NMR) spectroscopy. The three-cycle biosynthesis of BNC with an increasing volume of the nutrient medium from 10 to 72 L was successfully scaled up in nonsterile conditions to afford 9432 g of BNC gel-films. CNs with a nitrogen content of 10.96% and a viscosity of 916 cP were synthesized. It was found by the SEM technique that the CN preserved the 3D reticulate structure of initial BNC fibers a marginal thickening of the nanofibers themselves. Different analytical techniques reliably proved the resultant nitration product to be CN. When dissolved in acetone, the CN was found to form a clear high-viscosity organogel whose further studies will broaden application fields of the modified BNC.

## 1. Introduction

Research and advances in the field of bacterial nanocellulose (BNC) have been gaining fast pace in recent times, indicating the significance of this unique biopolymer. BNC has received global interest because of its physicomechanical and chemical properties, such as 3D porous structure, high purity, high mechanical strength and flexibility, high crystallinity, high specific surface area, high water-holding capacity and polymerization degree, as well as perfect biocompatibility [1]. 

Both as an individual material and in BNC-based composites, BNC has numerous applications in medicine (wound-healing materials, artificial skin, artificial blood vessels, tissue engineering biomaterials) [2,3,4,5], food industry [6], electronic industry [7], textile industry [8], paper industry, cosmetics and so forth [1,9]. Different modifications of BNC can expand its use to even more fields of promising applications. As any other cellulose, BNC can also be carboxymethylated, acetylated, phosphorylated, and modified by other reactions to produce a variety of BNC derivatives [10]. That being said, these derivatives will also exhibit unique properties, as they are derived from unique BNC. The promising direction in chemical modification of BNC is its nitration. Cellulose nitrates (CNs), owing to their unique physical properties, are among most essential cellulose derivatives widely used for industrial and defense purposes [11,12,13]. The synthesis of CNs from BNC will afford CNs with novel functionalities and properties, and will broaden the range of their application [10].

Initial results on the nitration of BNC are currently available. Yamamoto et al. [14] described the nitration of BNC with concentrated nitric acid in dichloromethane at 4 °C by varying the nitrating mixture composition and nitration time. Other researchers [11,15,16] obtained CNs by nitrating BNC with mixed nitric-sulfuric acid. The absence of a detailed description and the small quantity of the resultant CNs are the bottlenecks of the listed studies, which does not allow their full characterization. Luo et al. [11] prepared CNs from bacterial cellulose with a nitrogen content of 11.97−12.88%, and those authors believe the CNs have a rigid molecular chain in a dilute acetone solution. These findings can be used in further investigation of rheological and technological properties of BNC-based CNs and in search of their application field.

The literature overview on BNC nitration has revealed that full characterization of the CNs synthesized from BNC is absent, and the bacterial cellulose nitration issue is understudied. This is quite natural because BNC has moisture of 98−99%, while nitration requires dry BNC. The functionalization of the BNC surface makes sense only when the BNC biosynthetic process is systemized through to the production in required volumes [17]. Correspondingly, to achieve representative results and run necessary nitration replicates, it is required at the least that the BNC biosynthesis process be scaled up in volume. 

Individual strains are chiefly utilized as the BNC-producing microorganism. However, such microorganisms are distinguished by a spontaneous decline in cellulose-synthesizing capability and by an appreciable decrease in productivity in the manufacturing environment [18,19]. Therefore, a number of scientific teams put forward a concept of using microbial consortia whose adaptivity is enhanced by synergistic effects in the total metabolism [1], which is especially crucial for alternative nutrient broths prepared from residues of existing food, textile and hydrolytic industries, or prepared from worthless cellulosic raw materials [20]. The peculiar feature of the present study is that a *Medusomyces gisevii* symbiont was employed, also known as kombucha (tea fungus) [21]. Since this cellulose-producing symbiont exhibits adaptivity and capability of functioning under extreme conditions (for example, in heavy water) [22] and is tolerant to contamination, we have made an assumption that it can function in nonsterile conditions. The demand for such cellulose-producing strains was mentioned in the world literature [23]. In this case, BNC is synthesized for further use in technical chemistry to obtain a derivative, such as cellulose nitrate (CN); therefore, the possibility of working in nonsterile conditions is extremely important because it allows one not only to set up sustainable manufacture, but also reduce operational costs.

We have previously shown that BNC produced by *Medusomyces gisevii* has a high crystallinity index of 86−93% and is composed of 93.6−100% Iα-phase [24,25]. The triclinic cellulose Iα is less stable than the monoclinic cellulose Iβ, and hence, it has a higher reactivity; therefore, the Iα-phase will be the site of primary reaction, which is essential for derivatization. This is another more argument for choosing the symbiont.

The present study was aimed at producing an enlarged BNC sample and synthesizing CN therefrom. 

## 2. Materials and Methods 

All the reagents and materials used in this study were procured from AO Vekton, Russia. 

### 2.1. Biosynthesis of BNC

#### 2.1.1. BNC-Producing Symbiont

*Medusomyces gisevii* Sa-12, a BNC-producing strain, was procured from the All-Russian Collection of Industrial Microorganisms (State Research Institute of Genetics and Selection of Industrial Microorganisms of the National Research Center “Kurchatov Institute”, Moscow, Russia). The symbiont was maintained by the subculture method in a semisynthetic nutrient broth consisting of glucose (20 g/L) and black tea extract (5 g/L) [26], which is a conventional medium for a symbiont [27]. The conditions were as follows: The medium was 1 L in volume; the strain was seeded once a week; the temperature was 28 °C; stationary culture; and the physiological condition of the symbiont was controlled by microscopy. The active acidity was not controlled intentionally, as any intervention adversely affects the BNC-producing capability of the symbiont [26,28]. 

#### 2.1.2. Synthesis of BNC Gel-Films 

BNC was prepared in a cyclic manner whereby the nutrient medium was increasing in volume with each subsequent culturing. The variable culture parameters are summarized in Table 1. 

The culture in each cycle was run under stationary conditions for seven days at 20−24 °C in vessels made of stainless steel. The inoculum dosage in each cycle was 10% of the nutrient medium volume, in which case the growth medium obtained in the first culture cycle was used as the inoculum for the second cycle, and the growth medium from the second cycle was used as the inoculum for the third cycle.

The growth medium was analyzed for glucose concentration by UNICO UV-2804 spectrophotometer (United Products and Instruments, Inc., USA) using 3,5-dinitrosallycilic acid as the reagent (Panreac, Spain) [29]. The active acidity level of the culture medium was measured potentiometrically by an I-160MI ion meter (OOO Izmeritelnaya Tekhnika, Russia). The microbial abundance was analyzed by microscopy of the growth medium samples on an Optika B-150 instrument (Optika, Italy). The cell count was done in a Goryaev chamber. 

#### 2.1.3. Washing of BNC 

The passive purification technique was used because it can preserve the BNC structure to the full. The passive purification implies passive diffusion of dilute alkaline and acid solutions and distilled water, that is, the holding of BNC in the respective solutions for a prolonged time at 20−24 °C without stirring. The washing intensification (by increasing the alkali concentration or temperature) is known to bring about possible alterations in the structure of bacterial cellulose [30,31]. Here, BNC was washed in a 2% NaOH solution at 20−24 °C, and since the BNC gel-films had a large weight and thickness, the washing solutions were replaced 2−3 times. Then, BNC was washed with water to decolor BNC completely. Afterwards, decationation with a 0.1% HCl solution was performed, and BNC was then washed with distilled water until neutral wash waters.

#### 2.1.4. Calculation of BNC Yield 

The yield of BNC (%) was calculated by the Equation (1):(1)$$\eta =\frac{m_{BNC}}{C_{g}\cdot V\cdot 0.9}\cdot 100,$$where *η* is the BNC yield (%), *m_BNC_* is the weight of the BNC sample on an oven-dry basis (g), *C**_g_* is the glucose concentration in the medium (g/L), *V* is the volume of the medium (L), and 0.9 is the conversion factor attributed to the detachment of the water molecule by polymerization of glucose into cellulose. This calculation method is a modification of classical Hestrin and Schramm’s calculations [32] widely used in the biosynthesis of BNC [33]. The factor of 0.9 should be taken into account from the standpoint of stoichiometry of the BNC biosynthesis from glucose molecules. 

#### 2.1.5. Quality Attributes of BNC 

The α-cellulose content in the BNC was determined by a method whereby cellulose is treated with a 17.5 wt% NaOH solution (45 mL) for 45 min, and the undissolved residue is quantified after washing with 9.5 wt% NaOH and water, and then dried [34].

BNC degree of polymerization (DP) was determined by the outflow time of cellulose solution in cadoxene (cadmium oxide in ethylenediamine) from a VPZh-3 capillary viscometer (Ecokhim, Russia) with a capillary diameter of 0.92 mm [35].

All experiments were done in triplicate and data were expressed as average values. 

#### 2.1.6. Examination of BNC Structure 

The surface morphology of BNC fibers was examined by scanning electron microscopy (SEM) on a GSM-840 electron microscope (Jeol, Japan) after sputter-coating a Pt layer of 1−5 nm thick. The diameter of microfibrils was defined as an average of 50 measurements in the SEM image at ×10,000 zoom.

Infrared (IR) spectra of BNC were taken on an Infralum FT-801 spectrometer (OOO NPF Lumex-Sibir, Russia) operating at 4000−500 cm^−1^. For IR spectroscopy, BNC was pressed into pellets with potassium bromide in a BNC:KBr ratio of 1:150. 

#### 2.1.7. Freeze-Drying of BNC

BNC was freeze-dried in a Bio-Rus-4SFD (distributed by OOO Bio-Rus, Russia) freeze-dryer as follows: BNC was frozen at −40 to −50 °C and held at this temperature for 12 h, then BNC was gradually heated from −50 °C to +25 °C for 24 h. 

### 2.2. Nitration of BNC 

#### 2.2.1. Preparation of BNC for Nitration

Prior to nitration, the BNC freeze-dried to moisture of at most 5% was cut with scissors into 3 mm × 5 mm rhombic pieces. 

#### 2.2.2. Nitration and Stabilization of BNC 

BNC was nitrated under conditions required for the synthesis of low-nitrogen CN. A weighed portion of BNC (20 g) was treated with commercial sulfuric-nitric mixed acid containing 14% water. The mass ratio of BNC to mixed acid was 1:50, the temperature was 25−30 °C, and nitration time was 40 min. The CN sample washed until neutral wash waters were stabilized with constant stirring as follows: Treatment with water for 1 h at 85−95 °C, then treatment with 0.03% sodium carbonate solution for 3 h at 85−95 °C, and then again with water for 1 h at 85−95 °C. 

The CN sample dried at 100 ± 5 °C was analyzed. All experiments were done in triplicate and data were expressed as average values. 

#### 2.2.3. Calculation of CN Yield 

The yield of CN (%) was calculated by the Equation (2): (2)$$W=\frac{m_{CN}}{m_{BNC}}\cdot 100,$$where *W* is the yield of CN (%), *m_CN_* is the weight of the CN sample (g), and *m_BNC_* is the weight of the BNC sample for nitration (g).

The degree of substitution was calculated from the nitrogen content by Equation (3) [36]: (3)$$DS=\frac{3.6\;\left[N\right]}{31.1-\left[N\right]}.$$

#### 2.2.4. Analysis of CN 

The nitrogen content was quantified by the ferrous sulfate method [37,38], which relies on saponifying CN with concentrated sulfuric acid and on reducing the formed nitric acid with iron (II) sulfate to nitrogen oxide that generates, in excess of iron (II) sulfate, a [Fe(NO)]SO_4_ complex compound that turns the solution yellow-pink. The CN viscosity was determined by measuring the flow time of a 2% CN-acetone solution out of a VPZh-1 capillary viscometer (Ecokhim, Russia). The CN solubility in the alcohol-ester mixture was measured by filtering the CN residue insoluble in alcohol-ester mixture, followed by drying and weighing. The solubility in acetone (1 g CN and 50 mL acetone) was measured by filtration of the acetone-insoluble CN residue, followed by drying and weighing. The ash content was quantified by slowly decomposing CN with concentrated nitric acid upon heating, followed by incinerating and weighing the calcined residue.

#### 2.2.5. Examination of CN Structure

The surface morphology of CN fibers was studied, and IR spectroscopy of CN was performed by the same methods as for BNC.

Combined thermogravimetric (TGA) and differential thermal (DTA) analyses of CN were done on a TGA/DTG-60 thermal analyzer (Shimadzu, Japan) under the following conditions—0.5 g sample weight, 10 °C/min heating rate, 350 °C maximum temperature, and nitrogen as the inert environment. 

^13^C nuclear magnetic resonance (NMR) spectra of CN were recorded at 60 °C on a Bruker AM 400 spectrometer (Bruker, Germany) operating at 100.61 MHz. The chemical shifts of signals were referenced to DMSO-d_6_ as an internal standard. 

## 3. Results and Discussion

### 3.1. Biosynthesis of BNC

The scaled up biosynthesis of BNC afforded 9432 g of BNC gel-films. Table 2 lists basic indicators of BNC biosynthesis by the *Medusomyces gisevii* Sa-12 symbiont in three cycles. 

The BNC yield varied from 4.9% to 6.4% in three cycles. Because only the configuration of the vessels was varied, while the other parameters were kept unchanged, it can be inferred that the BNC yield depends on the thickness of the growth medium layer, that is, the thinner the layer of the medium, the higher the BNC yield: The thinnest layer of 2.9 cm in cycle I; a thicker layer of 3.7 cm in cycle III; and a yet thicker layer of 4.9 cm in cycle II (Table 1). 

By comparing the yield of BNC prepared in vessels of 17−104 L in volume with the lab-scale yield of 6.8% obtained in the 0.5–L vessel [26], one can state that the yield declined upon scaling up the biosynthesis of BNC. Despite the decrease in BNC yield during scale-up, a positive aspect was the absence of a relationship between the BNC yield and the number of cycles: For instance, the yield in the third cycle was higher than that in the second. Thus, the triple cyclic culturing with *Medusomyces gisevii* Sa-12 in nonsterile conditions was a success. The engineering aspects of this strategy should be further elaborated in order to obtain a high, stable yield of BNC with increasing the number of cycles. 

The change in glucose concentration during the biosynthesis of BNC is depicted in Figure 1. 

The residual glucose concentration indirectly indicated that the BNC yield varied in inverse proportion: The less glucose was left, the greater the BNC yield. The most representative sample of BNC weighing 6800 g and exhibiting an α-cellulose content of 99% and a polymerization degree of 4000 was used for nitration. 

### 3.2. Nitration of BNC

The key characteristics of the resultant CN are given in Table 3. 

The obtained CN sample had the following quality attributes—10.96% nitrogen content, 916 cP viscosity, 47% solubility in the alcohol-ester mixture, and 0.10% ash content. The solubility test of CN in acetone showed a 100% result. 

The yield of CN was calculated as the weight of initial cellulose and was 158%, which is due to the increase in the average molecular weight of the monomeric unit by incorporation of the nitro group.

The small nitrogen content of 10.96% can be attributed to the low reactivity of BNC, due to its lightweight. The lightweight rhombic pieces of BNC had a volumetric shape and were recalcitrant to impregnation with mixed acid, floating on the surface of the reaction mixture in spite of agitation. Thus, the diffusion of the nitronium cation into the BNC sample slowed down, in a similar manner described [13]. 

The findings differ significantly from those obtained for nitration of conventional plant cellulose. Conspicuous is the fact that the resultant BNC-based CN sample had a higher viscosity of 916 cP versus 0.6−72 cP of industrial CNs [36] and versus 4−35 cP of plant CNs from *Miscanthus* [38,39], intermediate flax straw [40] and oat hulls [41], which were produced under similar conditions.

No doubt that the viscosity of BNC-based CN can be reduced to the required values by autoclaving under conditions harsher than those for plant cellulose-based CN [38,39,40,41]. The viscosity reduction will increase the solubility of BNC-based CN. 

The substantial difference between characteristics of BNC-based CN and plant cellulose-based CN is due to the unique 3D reticulate structure, high degree of polymerization, and plate-like shape of BNC. Taking cognizance of the features of BNC-based CN, one should seek special fields of application in which high-viscosity properties of CN are required. It was found by measuring the viscosity (dissolution of 1 g CN in 50 mL acetone) that BNC-based CN generated a transparent, highly viscous organogel that represents a continuous 3D macromolecular network serving as a framework whose voids are filled with low-molecular acetone. The potential application fields of this organogel have not yet been identified. Probably, the high purity and unique structure of BNC-based CN will allow it to be used in modern science-driven areas, for example, in adhesive formulations for gluing items and electronics components, in special-purpose nitrolacquers, etc.

### 3.3. Analysis Results for BNC and CN Samples 

SEM images of BNC and BNC-based CN are displayed in Figure 2, showing that the reticulate structure of original BNC fibers retained during nitration. 

The average diameter of microfibrils of original BNC was 97 nm (a mean-square deviation of 21 nm). The average diameter of CN fibers was 114 nm (a mean-square deviation of 22 nm). Thus, as opposed to plant cellulose [41], the nitration of BNC almost did not enlarge the diameter of CN fibers.

Figure 3 shows the IR spectra of original BNC and BNC-based CN. 

The IR spectra of BNC and CN show the basic characteristic frequencies of cellulose and CN [36,42,43,44,45]. The IR spectrum of original BNC had an intensive absorption band near 3419 cm^−1^, indicating OH-group stretching. The IR spectrum of CN had absorption bands distinguishing it from that of initial BNC. In the region of the OH-group (3500 cm^−1^), the IR spectrum of CN exhibited a much lower area compared to that of original BNC, suggesting only a partial substitution of OH-groups by NO_2_, which is in good agreement with the low nitrogen content of the resultant CN (10.96%). Besides, the following characteristic frequencies were detected in the IR spectrum of CN—2555, 1655, 1280, 834, 743 and 681 cm^−1^, which match the basic absorption bands of nitro groups [36,42]. Similar stretch vibrations were also found in the IR spectrum of BNC-based CN by Sun et al. [15]. The characteristic frequency of decomposition products of CN at 2300 cm^−1^ was missing. 

In addition, the CN sample was evaluated by TGA/DTA and ^13^C NMR spectroscopy. The TGA/DTA scans of the BNC-based CN sample are displayed in Figure 4. 

The TGA/DTA analysis showed that the resultant CN sample had a high decomposition temperature. The TGA curve had one exothermic peak at 210 °C, accompanied by the sample weight loss to 83%. The onset temperature of decomposition of the synthesized CN sample was 200 °C and extended to 220 °C. The obtained results are on a par with the experimental data for TGA/DTA of BNC-based CN [15].

The ^13^C NMR spectrum of the BNC-based CN sample is depicted in Figure 5. 

The NMR spectrum of BNC-based CN showed chemical shifts typical of 6-mononitrocellulose (76.6 ppm), 2,6-dinitrocellulose (97.8 ppm, 84.3 ppm, 82.9 ppm, 79.4 ppm), 3,6-dinitrocellulose (78.2 ppm) and 2,3,6-trinitrocellulose (99.2 ppm, 73.6 ppm, 70.9 ppm) [46,47]. Overall, the CN samples contained 6-, 2,6-, 3,6- and 2,3,6-substituted moieties of the glucopyranose ring of the cellulose macromolecule. These findings match the results earlier obtained for the nitration of bacterial cellulose with mixed anhydrous HNO_3_ and dichloromethane [14].

The obtained results from our studies on the properties of CN derived from microbial cellulose suggest that new CNs can potentially be used in conventional and high-tech economic sectors. But these efforts can be justified only by scaling up the biosynthesis of BNC with stable characteristics of BNC from worthless non-food plant biomass using a cellulose-producing strain with a high-level adaptation and a high tolerance to contamination [48], which is being planned to be done in the future. 

## 4. Conclusions

The three-cycle biosynthesis of BNC by the *Medusomyces gisevii* Sa-12 symbiont with increasing volume of the nutrient medium from 10 to 72 L was successfully scaled up in nonsterile conditions to furnish 9432 g of BNC gel-films. The 6800-g BNC sample chosen for nitration exhibited an α-cellulose content of 99% and a polymerization degree of 4000. CNs having a nitrogen content of 10.96% and a viscosity of 916 cP were synthesized by nitration of freeze-dried BNC with sulfuric-nitric mixed acid. It was found by comparing the surface morphologies of BNC and CN fibers that the 3D reticulate structure of original BNC retained in full with the nanofibers themselves thickened marginally. Different techniques, such as IR, TGA/DTA, and ^13^C NMR, verified that the resultant nitration product was CN. The CN, when dissolved in acetone, was found to generate a clear high-viscosity organogel whose further studies will likely open up new application fields of CN. 

## Figures and Tables

**Figure 1 nanomaterials-09-01694-f001:**
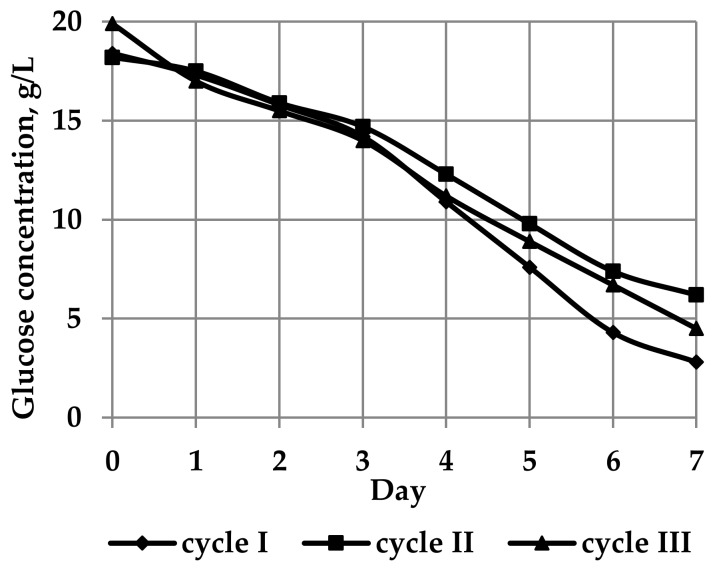
A time profile of glucose concentration during biosynthesis of BNC.

**Figure 2 nanomaterials-09-01694-f002:**
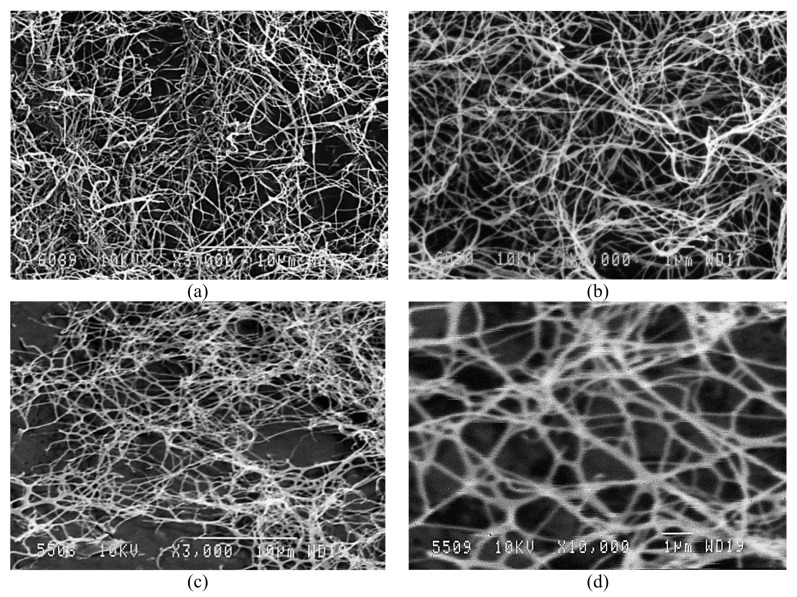
Scanning electron microscopy (SEM) images of (**a**,**b**) BNC and (**c**,**d**) BNC-based CN at zooms ×3000 and ×10,000, respectively.

**Figure 3 nanomaterials-09-01694-f003:**
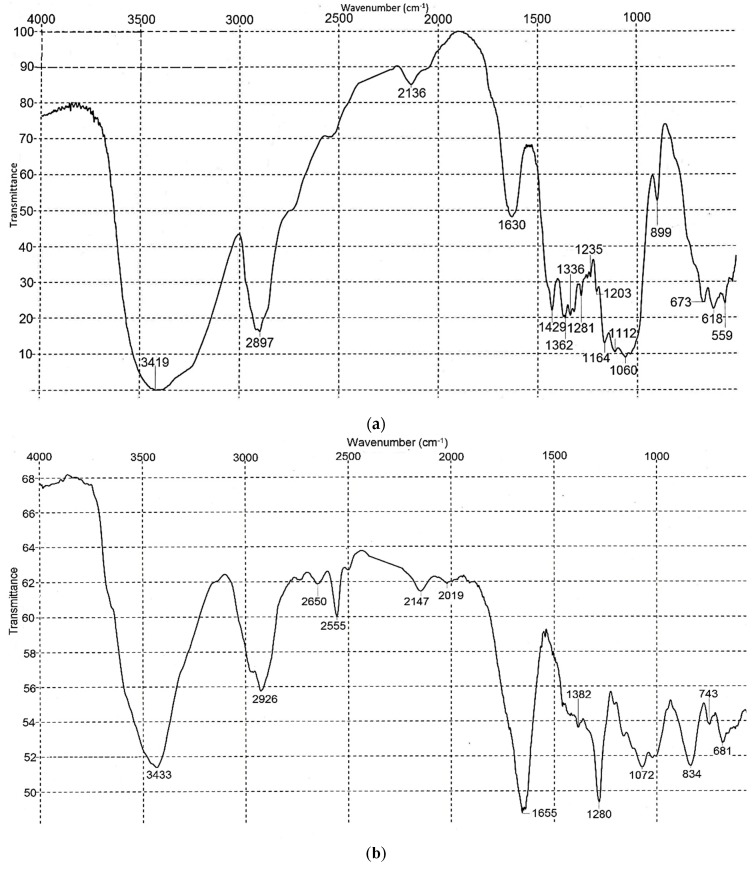
Infrared (IR) spectra of (**a**) BNC and (**b**) CN.

**Figure 4 nanomaterials-09-01694-f004:**
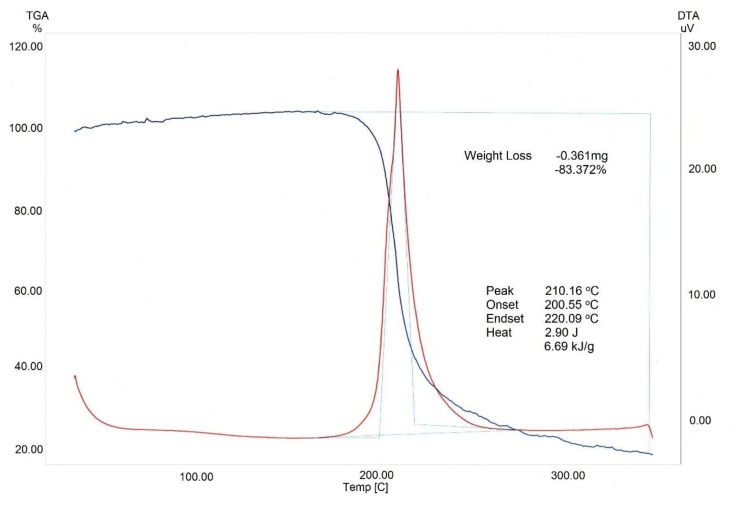
Thermogravimetric (TGA) and differential thermal (DTA) pattern of BNC-based CN.

**Figure 5 nanomaterials-09-01694-f005:**
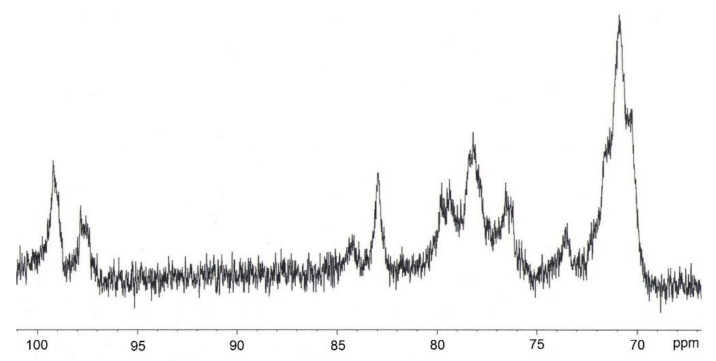
^13^C nuclear magnetic resonance (NMR) spectrum (DMSO-d_6_) of BNC-based CN.

**Table 1 nanomaterials-09-01694-t001:** Cyclic culture of bacterial nanocellulose (BNC).

Culture Cycle	Culture Vessel Dimensions (cm × cm × cm)	Culture Vessel Volume (L)	Fullness Coefficient of Culture Vessel	Qnty of Vessels (pcs)	Total Volume of Nutrient Medium (L)	Thickness of Medium Layer (cm)
I	49.5 × 69.5 × 5.0	17	0.59	1	10	2.9
II	36.5 × 36.5 × 8.3	11	0.59	3	19.5	4.9
III	140 × 140 × 5.3	104	0.69	1	72	3.7

**Table 2 nanomaterials-09-01694-t002:** Basic indicators of BNC biosynthesis.

Culture Cycle	Total Volume of Medium (*V*, L)	Residual Volume of Medium (L)	Initial Glucose Concentration in Medium (*C_g_*, g/L)	Residual Glucose Concentration in Medium (g/L)	BNC Gel-Film Weight (*m*, g)	BNC Yield (*η*, %)
I	10.0	4.3	18.4	2.8	1074	6.4
II	19.5	11.9	18.2	6.2	1558	4.9
III	72.0	42.8	19.9	4.5	6800	5.2

**Table 3 nanomaterials-09-01694-t003:** Key properties of synthesized cellulose nitrate (CN).

Sample Name	Nitrogen Content (%)	Viscosity of 2% CN-Acetone Solution (sP)	Solubility in the Alcohol-Ester Mixture (%)	Ash Content (%)
BNC-based CN	10.96	916	47	0.1

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
