# Peer review of "Bacterial Nanocellulose Nitrates"

_nanomaterials, 2019, doi:10.3390/nano9121694_

Round 1

Reviewer 1 Report

Interesting article, comments below:

What is the viscosity of the starting BNC before nitration? How can you assert that the higher viscosity observed for the nitrated BNC is not due to unreacted material? Regarding viscosity, comparing your sample with commercial Cellulose nitrates is not correct. The procedure to make each differs. In this sense for the nitration experiment your experimental design is lacking a control sample of plant cellulose to compare with. It makes sense to have nitrated a sample under same conditions and determine its viscosity to compare. Similarly, for the IR curves, and the TGA analysis, the nitration of BNC can be compared to that of plant cellulose. The accessibility of the samples is different, and this affects the nitration yield. SEM images in figure 2 can be improved, higher magnification images look very blurry. Curious to know if the organogel behavior observed for the nitrated sample could be also observed for the BNC non-nitrated sample. Is it happening because of a unique feature of the nitrated BNC or is it actually because of the fibrillar structure of BNC?

Author Response

Please see the Pdf attachment.

Reviewer 2 Report

The manuscript by Budaeva and coworkers describes the large scale production of bacterial nanocellulose (BNC) and the subsequent nitration and characterization of the product. The main findings include the identification of conditions compatible with a larger scale production and the finding that the cellulose nitrate (CN) produced forms organogels with much higher viscosity than those from other cellulose sources. The paper would be stronger if some of the possible applications had been demonstrated since this would provide added justification for producing CN from BNC. Nevertheless there is sufficient new material for publication of the manuscript after the following points have been dealt with.

The introduction (lines 47-8) states that cellulose nitrates are important due to their unique properties. It would be useful to provide some information on what properties are unique. Lines 61-62 note that BNC has a moisture content of 98-99%. Does that mean that the water content is approximately the same as the cellulose content? A reference is missing for the sentence starting at line 81 (previous results on BNC production). The sentence on lines 86-87 is somewhat unclear. I presume the authors mean a large scale production of a BNC sample. It would be helpful to explain what is meant by an enlarged sample. A brief explanation of what is meant by the passive purification method should be added to Section 2.1.3. Table 3 does not provide adequate information to understand the solubility of CN. The paragraph on the nitrogen content (lines 232-238) is unclear. The CN prepared here has a nitrogen content of ~ 11%, similar to the 12% value cited for plant cellulose. It is not clear how the authors can conclude whether the nitration is homogenous, simply by comparing these two values. Lines 242-244 state that the viscosity of BNC-based CN can be reduced by autoclaving. Has this experiment been done?   And why would it be desirable to do this since the authors later claim that the high-viscosity CN may lead to new applications? The images in Figure 3 look as though the individual microfibrils are larger and more variable in size than the values cited in lines 260-263 for both BNC and CN. What are the quoted sizes based on? How many fibrils are measured to determine this number? The presentation of Figure 3 should be improved. The numbers that identify specific peaks and the axis labels are too small to read. The caption for Figure 5 should provide the solvent for the NMR spectrum. The peak assignments should be provided in the text.

Author Response

Please see the Pdf attachment.

Round 2

Reviewer 1 Report

Including the references about nitrated plant cellulose samples under same conditions help supports your claims.

 My recommendation is that if these nitrated plant cellulose samples were available, the IR and TGA curves of these samples would give stronger support to your manuscript (at least one) as you are trying to prove applicability on industrial scale. 

Reviewer 2 Report

The authors have dealt satisfactorily with my suggestions and questions on their original manuscript.
